# Triggering Generative Collapse: A Contrastive Inversion Framework for AI-Generated Image Detection

## Abstract

With the rapid evolution of generative models, AI-generated synthetic content has become increasingly realistic and difficult to detect, posing serious threats to information integrity. Unlike authentic samples that align with well-formed data distributions, manipulated samples often fall off the underlying data manifold. The foundational insight of our work is that a diffusion model's response to such off-manifold inputs can be deliberately engineered. We demonstrate that through targeted contrastive fine-tuning, subtle latent deviations consistently result in structured reconstructions failures. Leveraging this engineered sensitivity, we introduce the Contrastive Reconstruction Amplification Forensics Technique (CRAFT), a method that actively induces generative collapse for forgeries while maintaining faithful reconstruction for authentic content. The core of our approach is a contrastive reconstruction loss that fine-tunes the terminal stages of the DDIM denoising process. This optimization fundamentally alters the model's dynamics, engineering a deliberately fragile and asymmetric generative manifold: authentic latents are guided toward high-fidelity reconstructions, while any latent perceived as off-manifold is aggressively pushed toward a state of collapse. To further amplify this divergence, our framework then employs a class-guided latent perturbation that pushes all inputs away from the real-class center, effectively steering forged latents into the "failure regions" engineered by our fine-tuning. The result is a predictable structural collapse for forged samples into distinctive artifacts (e.g., honeycomb patterns), exposing a failure mode rooted in the model's divergent Jacobian dynamics. This induced collapse yields a strong and interpretable authenticity cue that enables robust detection. Extensive experiments demonstrate that our method achieves superior performance and cross-domain generalization.

## 1 Introduction

Modern GANs and diffusion models can now synthesize content virtually indistinguishable from reality Goodfellow et al. (2020); Rombach et al. (2021); Karras et al. (2021a); Ho et al. (2020); Bhattad et al. (2024). While such progress drives innovation in creative industries, simulation, and data augmentation, it also raises urgent concerns about authenticity and trust. AI-generated content can be easily misused for misinformation, identity fraud, copyright infringement, and political manipulation, making reliable detection a critical and growing challenge Corvi et al. (2023).

As synthetic imagery becomes increasingly photorealistic, detectors that rely on visual artifacts, frequency anomalies, or temporal cues are steadily losing effectiveness Li & Lyu (2019); Afchar et al. (2018); Qian et al. (2020); Rajan et al. (2025); Nirkin et al. (2022); Huang et al. (2023). A recent counter-trend, reconstruction-based detection, assumes generators better reproduce authentic images and thus treats reconstruction discrepancies as a cue Wang et al. (2023a); Ricker et al. (2024). Yet this paradigm is inherently passive: expressive generators often recreate both real and fake inputs with high fidelity, resulting in small or ambiguous errors and dilute discriminative power Ojha et al. (2023a); Lim et al. (2024).

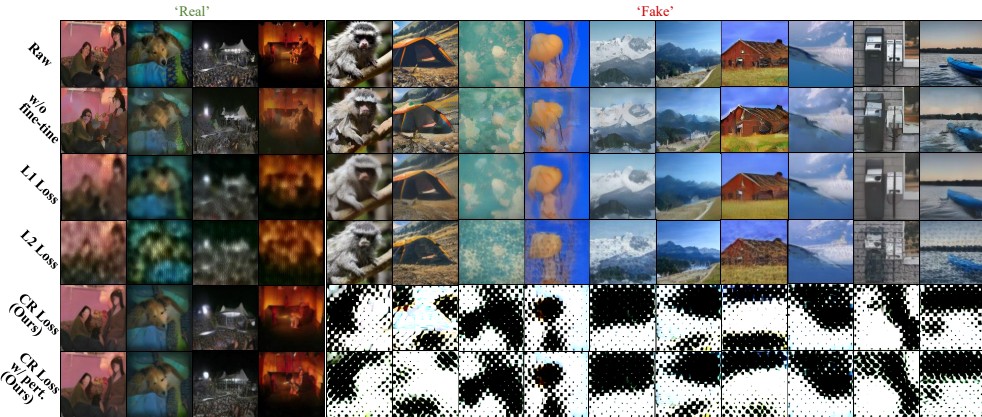

Figure 1: **Reconstruction results under different settings.** Top to bottom: input images, reconstruction without fine-tuning or perturbation, fine-tuned with L1 loss, L2 loss, CR loss, and CR loss with latent perturbation. Without fine-tuning, reconstruction lacks class-discriminative signals. L1 and L2 loss introduces moderate differences but degrades real samples. In contrast, our CR loss preserves authentic details while collapsing forged reconstructions into structured artifacts. Perturbation further amplifies this asymmetry.

This work proposes a fundamental shift. Rather than passively evaluating reconstruction behavior, we ask: *Can the generative model itself be provoked into revealing a fake?* We answer in the affirmative by introducing an active-induced failure paradigm that deliberately drives the generator into a structured collapse, but only for manipulated content. The key lies in exploiting the latent geometry of diffusion models, turning detection from passive observation into active provocation.

We instantiate this idea as the Contrastive Reconstruction Amplification Forensics Technique(CRAFT), which engineers divergent generative pathways between authentic and synthetic samples. The core of our approach is a stage-wise fine-tuning strategy applied to the final DDIM denoising steps. This instills a fragile yet asymmetric generative dynamic: forged latents collapse, while authentic ones remain robust. To further magnify this divergence, we apply a class-guided latent perturbation that shifts all inputs away from the authentic-class center. This pushes forgeries deeper into off-manifold regions, making them more prone to failure.

This interaction between our components triggers a cascading amplification process: (1) our fine-tuning first establishes a fragile, asymmetric dynamic within the terminal denoising steps, making the model hypersensitive to off-manifold inputs, (2) the class-guided perturbation drives forged latents into unstable regions of the latent space, initiating a cascade of divergence, (3) finally, the VAE decoder, unable to interpret this pathological input, undergoes a structural collapse, manifesting as the distinctive honeycomb artifacts that serve as interpretable indicators of inauthenticity.

Rather than relying on subtle discrepancies, our framework activates the model's intrinsic sensitivity to synthetic content, turning the denoising process itself into a forensic tool. By integrating latent-space geometry, contrastive supervision, and denoising sensitivity, our method achieves robust, interpretable, and highly generalizable detection of AI-generated images across semantically and visually distinct domains. Our main contributions are summarized as follows.

- We introduce a novel active failure induction paradigm for diffusion models, transforming detection from passive error measurement into active structural interrogation of generative collapse.

- We design a class-guided latent perturbation that shifts all samples away from the authentic-class center, preserving real inputs near the manifold boundary while pushing synthetic forgeries deeper into off-manifold regions to exacerbate instability.

- We formulate a contrastive reconstruction loss that exploits the chaotic cascade of DDIM denoising to enforce reconstruction asymmetry: authentic samples are reconstructed faithfully, while forgeries are driven toward collapse with distinctive, human-interpretable artifacts.

- We implement a stage-wise adaptation strategy on the terminal DDIM denoising steps, converting off-manifold chaos into a learned failure mode, delivering strong cross-domain generalization with minimal additional compute.

## 2 RELATED WORK

The rise of photorealistic generative models has spurred the development of forgery detection methods. Early efforts rely on visual artifacts, frequency inconsistencies, or physiological signal anomalies, while more recent work shifts toward reconstruction-based strategies that leverage pretrained generative models to measure fidelity gaps between real and fake content.

### 2.1 TRADITIONAL FORGERY DETECTION

Classical approaches target surface-level cues such as artifacts Li & Lyu (2019); Afchar et al. (2018); Pellcier et al. (2024), frequency domain traces Qian et al. (2020); Lanzino et al. (2024), or physiological irregularities Agarwal et al. (2020); Qi et al. (2020). For instance, MesoNet Li & Lyu (2019) exploits warping artifacts from face generation pipelines, F3Net Qian et al. (2020) introduces frequency-aware learning using DCT decompositions, and DeepRhythm Qi et al. (2020) tracks remote PPG signals to uncover inconsistencies in heartbeat patterns. Though effective in narrow settings, these methods are fragile, as they are rooted in artifacts that can be easily masked by generative advancements and offering limited insight into distribution-level deviations.

### 2.2 RECONSTRUCTION-BASED DETECTION

To overcome the limitations of traditional approaches, recent efforts have shifted toward reconstruction-based methodsCao et al. (2022), which leverage the generative behavior of pretrained models. These methods typically reconstruct an input image and compute its discrepancy from the original Ricker et al. (2024). This based on the premise that authentic images, lying closer to the learned data manifold, can be more faithfully recovered than forged ones, which deviate from the training distribution and thus exhibit higher reconstruction errors. This line of research offers a model-aligned and semantically meaningful alternative to conventional classification-based detection. Building on these insights, Ojha et al. Ojha et al. (2023a) proposed a training-free detection paradigm utilizing pre-trained vision–language model features, demonstrating generalization across unseen generative models. Extending this concept, DIRE Wang et al. (2023a) introduces an image representation that quantifies the discrepancy between an input image and its reconstruction by a pre-trained diffusion model. The authors observed a critical asymmetry: real images can be more faithfully reconstructed than forged ones. This reconstruction gap serves as an effective signal for distinguishing real from generated content, enabling robust detection. Similarly, FakeInversion Cazenavette et al. (2024) leverages inversion-derived features from pre-trained Stable Diffusion models to train real-vs-fake classifiers and introduces a benchmark that better reflects practical deployment scenarios, thus demonstrating the utility of diffusion inversion as a source of discriminative representations.

Unlike prior work that passively observes reconstruction errors, we propose an active collapse induction strategy. By fine-tuning the terminal diffusion stages, we construct a hypersensitive generator that maintains stability for real inputs but collapses under forged ones. While class-guided perturbation amplifies this instability, our results show that fine-tuning is essential for triggering the collapse, transforming the generator from a neutral reconstructor into a forensic amplifier.

## 3 METHOD

In this section, we present our Contrastive Reconstruction Amplification Forensics Technique (CRAFT), a detection framework that transforms diffusion models into active forgery detectors by inducing asymmetric reconstruction dynamics. As illustrated in Figure 2. Our method leverages the deterministic reverse process of Denoising Diffusion Implicit Models (DDIM) to map an input image $x$ into its latent representation $z$. A class-guided perturbation then shifts $z$ away from the authentic-class center, producing a perturbed latent $z'$. This is decoded via the reverse diffusion process $\mathcal{D}_\theta$ to obtain the reconstruction $\hat{x}$. To differentiate real and fake inputs, We fine-tune the image

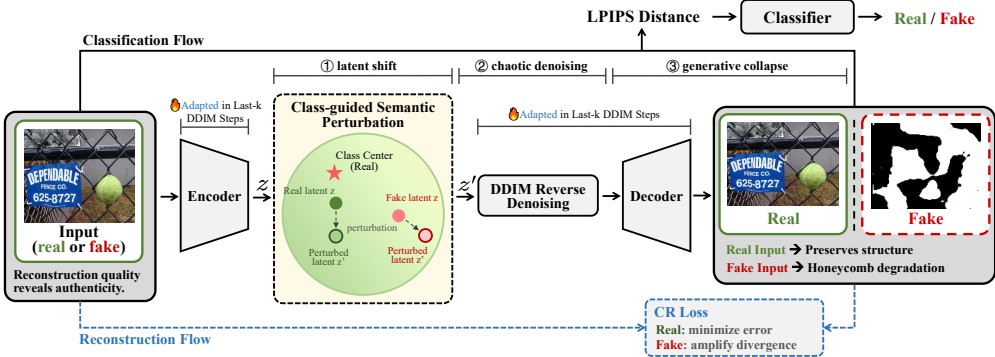

Figure 2: Overview of our proposed CRAFT. The framework comprises a Reconstruction Flow (bottom, for training) and a Classification Flow (top, for inference). Given an input image, we perform a (1) latent shift away from the authentic-class center. The resulting off-manifold latent then undergoes (2) chaotic denoising, where fine-tuning in the final DDIM steps amplifies instability. For forged samples, this process leads to a (3) generative collapse, producing structured artifact (e.g., honeycomb patterns) under the guidance of our Contrastive Reconstruction (CR) Loss. Authentic samples, by contrast, reconstruct faithfully. This stark asymmetry serves as a strong and interpretable cue for classification.

encoder, UNet, and VAE decoder specifically at the terminal steps of the DDIM denoising process using a contrastive reconstruction loss. This loss enforces stability for authentic inputs while amplifying collapse in forged ones, resulting in interpretable artifacts. Together, the perturbation and fine-tuning create an active failure pathway that reveals inauthentic content through generative failure.

## 3.1 LATENT INVERSION WITH DIFFUSION MODELS

We adopt DDIM inversion to reconstruct images from their latent representations. The Denoising Diffusion Implicit Model (DDIM) enables a deterministic reverse diffusion process that maps a noisy latent representation back to the image domain. The standard DDIM update from $x_t$ to $x_{t-1}$ is given by:

$$x_{t-1} = \sqrt{\alpha_{t-1}} \cdot \left( \frac{x_t - \sqrt{1 - \alpha_t} \cdot \epsilon_\theta(x_t, t)}{\sqrt{\alpha_t}} \right) + \sqrt{1 - \alpha_{t-1}} \cdot \epsilon_\theta(x_t, t) \tag{1}$$

where $\epsilon_\theta$ is the denoising UNet. For any input image x, we obtain its latent representation $z$ by first encoding it with the VAE and then applying a fixed number of forward diffusion steps. This latent $z$ serves as the starting point for our perturbation and reconstruction pipeline, denoted as $\hat{x} = \mathcal{D}_\theta(z)$.

## 3.2 LATENT PERTURBATION FROM REAL-CLASS CENTERS

To induce an initial separation between authentic and synthetic samples, we introduce a latent-space perturbation mechanism centered around the authentic data manifold. Specifically, We maintain a dynamic latent center $\mu_{\text{real}} \in \mathbb{R}^d$, computed as the exponential moving average over latent vectors $z$ of authentic training samples:

$$\mu_{\text{real}} = \frac{1}{|R|} \sum_{z_i \in R_c} z_i \tag{2}$$

$$R = \{ z_i \mid z_i \text{ is authentic} \} \tag{3}$$

Given an inverted latent vector $z \in \mathbb{R}^d$, we apply a perturbation from the real center to obtain the perturbed latent $z'$:

$$z' = z + \lambda \cdot \frac{z - \mu_{\text{real}}}{\|z - \mu_{\text{real}}\| + \epsilon} \tag{4}$$

Here $\lambda$ is a hyperparameter controlling the perturbation magnitude, and $\epsilon$ is a small constant for numerical stability. This class-agnostic perturbation has divergent effect: authentic latents, being close to $\mu_{\text{real}}$, are gently pushed toward the boundary of the data manifold, while forged latents, which are farther from the real center, are pushed deeper into off-manifold regions, where the model is more likely to collapse.

## 3.3 CONTRASTIVE FINE-TUNING FOR DIVERGENT RECONSTRUCTION

The core of our method is the Contrastive Reconstruction Loss ($\mathcal{L}_{CR}$), which fine-tunes the model to actively amplify the divergence created by the latent perturbation. The loss enforces two distinct behaviors based on the authenticity label $y$ ($y = 0$ for authentic, $y = 1$ for forged). The overall objective is to minimize $\mathcal{L}_{CR}$:

$$\min_\theta \mathbb{E}_{(x,y)}[\mathcal{L}_{CR}(x, \mathcal{D}_\theta(z'), y)] \tag{5}$$

Let $x$ be the input image and $\hat{x} = \mathcal{D}_\theta(z')$ be its reconstruction from the perturbed latent $z'$. Let $d = \text{diff}(x, \hat{x})$ denote a base distance (e.g., MSE or LPIPS) averaged spatially. The $\mathcal{L}_{CR}$ is defined as:

$$\mathcal{L}_{CR}(x, \hat{x}, y) = \begin{cases} d, & \text{if } y = 0 \\ \max(0, m - d) + \gamma \cdot \max(0, d - \epsilon_m), & \text{if } y = 1 \end{cases} \tag{6}$$

where $m$ is a lower bound that encourages the reconstruction error $d$ of forged samples to exceed a minimum degradation margin, $\epsilon_m = m + \delta$ defines a soft upper limit for acceptable degradation, introducing a tolerance window $[m, m + \delta]$ in which no additional penalty is applied. $\gamma$ controls the strength of penalization when $d$ exceeds $\epsilon_m$, thereby preventing collapse into overly noisy or degenerate outputs.

This two-sided design promotes structured degradation for forgeries. The first term enforces separation from real samples by increasing their reconstruction error, while the second term regularizes the collapse by steering the model toward a repeatable failure mode (e.g., honeycomb patterns) instead of trivial noise.

## 3.4 ARTIFACT-AWARE PERCEPTUAL LOSS AND TOTAL LOSS

To further guide the learning process, we add an auxiliary classification loss. A simple discriminator $f_{cls}$ (e.g., an MLP) takes the scalar LPIPS distance $d_{LPIPS}$ Zhang et al. (2018) as input and predicts the authenticity label. The classification loss $\mathcal{L}_{CE}$ is the standard cross-entropy loss on these predictions.

Let $x$ denote the original input image and $\hat{x}$ its reconstruction via the fine-tuned diffusion decoder. The LPIPS-based discrepancy, averaged spatially for each image, is defined as:

$$d_{LPIPS} = \text{Average}(\mathbf{LPIPS}_{spatial}(x, \hat{x})) \tag{7}$$

where $\mathbf{LPIPS}_{spatial}$ denotes the spatially resolved perceptual distance map, and $\text{Average}(\cdot)$ performs a mean operation across its spatial dimensions.

the classification loss $\mathcal{L}_{CE}$ is defined as:

$$\mathcal{L}_{CE} = -\mathbf{log}\left(\frac{\exp(\mathbf{s}_y)}{\sum_{k=0}^{1} \exp(\mathbf{s}_k)}\right) \tag{8}$$

where $\mathbf{s}$ is the logit vector output by $f_{cls}(dist)$, and $\mathbf{s}_y$ is the logit corresponding to the true class $y$.

Finally, the total objective used for end-to-end training is a weighted combination of the contrastive and classification losses:

$$\mathcal{L}_{total} = \mathcal{L}_{CR} + \lambda_{cls} \cdot \mathcal{L}_{CE} \tag{9}$$

where $\lambda_{cls}$ is a weighting hyperparameter. We jointly fine-tune the parameters $\theta$ of the diffusion backbone (Image Encoder, UNet, VAE Decoder) and the parameters of the discriminator $f_{cls}$ by minimizing $\mathcal{L}_{total}$. As detailed in our implementation, we employ differential learning rates, using very small rates for the pretrained backbone components while allowing the VAE Decoder and the discriminator to adapt more quickly, thereby efficiently learning the desired divergent behavior.

Figure 3: **Cross-domain generalization of CRAFT, trained solely on ImageNet.** We visualize reconstructions for authentic (left) and forged (right) samples across three distinct test domains. CRAFT consistently preserves fine details for authentic inputs while inducing a characteristic collapse for forgeries, regardless of whether the domain is in-distribution or out-of-distribution.

## 4  THEORETICAL ANALYSIS

The reverse process in DDIM is defined iteratively as $x_{t-1} = g_t(x_t; \epsilon_\theta(x_t, t, z^`))$, where the transition $g_t$ is a function of the denoising UNet, $\epsilon_\theta$. To characterize the stability of this process, we analyze its local local dynamics via Jacobian $J_t = \partial g_t / \partial x_t$. Our method engineers the behavior of this system to create a clear distinction between authentic and synthetic inputs.

### 4.1  CONTROLLED OFF-MANIFOLD PERTURBATION

Given an input latent $z$, we apply a category-guided perturbation to steer it to a new position $z'$. This is achieved by moving $z$ away from the authentic-class center $\mu_{\text{real}}$ along a normalized direction vector:

$$z' = z + \alpha \cdot \Delta_{\text{real}}, \quad \text{where} \quad \Delta_{\text{real}} = \frac{z - \mu_{\text{real}}}{|z - \mu_{\text{real}}|_2} \tag{10}$$

This operation has a divergent effect. For authentic latent near $\mu_{\text{real}}$, $z'$ remains on the latent manifold $\mathcal{M}_{\text{real}}$. However, a forged latent, already deviating from $\mathcal{M}_{real}$, is pushed further into a low-density, off-manifold region. To ensure a stable process, we constrain the resulting latent:

$$\|z'\|_2 \leq \tau \tag{11}$$

This yields a bounded but semantically disruptive initialization $x_T := z'$ for the reverse diffusion process.

### 4.2  CHAOTIC CASCADE DURING DENOISING

The stability of the denoising trajectory depends on the Jacobian norm $\|J_t\|$. When $x_t$ is on-manifold, $|J_t|$ remains bounded, yielding a smooth reconstruction. In contrast, an off-manifold $x_t$ leads to poorly calibrated predictions from $\epsilon_\theta$, especially in later steps as the noise diminishes and the latent's underlying structure is exposed.

This causes Jacobian norm to explode $\|J_t\| \gg 1$, triggering an exponential amplification of error between any two close trajectories:

$$\|x_{t-1} - x_{t-1}^*\| \approx \|J_t\| \cdot \|x_t - x_t^*\| \tag{12}$$

This recursive amplification drives the denoising process into a chaotic cascade. The final latent $x_0^{fake}$ is thus non-semantic, saturated with incoherent signals, and far from the authentic manifold $\mathcal{M}_{real}$. It is precisely this instability $\|J_t\| \gg 1$ that our Contrastive Reconstruction Loss trains the model to exhibit for forged inputs.

Table 1: **Evaluation of cross-domain generalization**. Each detection method is trained independently on a single source dataset and evaluated by **Accuracy(%)** across three overall test domains. Methods marked with * indicate seen (i.e., used during training). We abbreviate StyleGAN as SGAN and ProjectedGAN as PGAN for brevity.

| Method | Training Dataset | ImageNet | | | CelebA | | | LSUN | | | | | | | | |
|---|---|---|---|---|---|---|---|---|---|---|---|---|---|---|---|---|
| | | Total | ADM* | SDv1 | Total | SDv2* | IF | Total | SGAN* | PGAN | iDDPM* | SDv2 | LDM | VQD | IF | Mid. |
| CNN-Det | ImageNet | 82.67 | 98.28 | 75.76 | 75.22 | 91.80 | 63.80 | 70.40 | 92.30 | 43.20 | 77.80 | 99.20 | 86.30 | 98.60 | 99.20 | 98.00 |
| | CelebA | 35.45 | 6.540 | 20.28 | 71.00 | **100.0** | 50.70 | 24.90 | 18.30 | 14.30 | 15.20 | 84.10 | 38.10 | 14.00 | 24.00 | 0.000 |
| | LSUN | 64.15 | 75.34 | 46.38 | 74.72 | 93.40 | 33.80 | 92.61 | 99.10 | 86.10 | 99.80 | 98.10 | 98.00 | 98.20 | **100.0** | 56.00 |
| DIRE | ImageNet | 81.69 | 99.60 | 64.11 | 96.44 | 99.90 | 97.70 | 67.20 | 70.60 | 54.60 | 73.60 | 98.70 | **100.0** | 99.80 | **100.0** | 89.00 |
| | CelebA | 32.25 | 3.540 | 13.44 | 71.92 | **100.0** | 52.90 | 12.01 | 5.200 | 1.200 | 0.700 | 61.10 | 26.40 | 4.200 | 9.800 | 0.000 |
| | LSUN | 68.66 | 65.26 | 55.68 | **96.92** | **100.0** | 97.00 | 93.12 | **100.0** | 97.60 | **100.0** | 84.90 | **100.0** | 99.20 | 99.70 | 13.00 |
| AEROBLADE | training-free | **81.06** | 66.70 | **94.41** | 91.23 | 88.50 | **99.85** | 68.65 | 55.75 | 59.65 | 71.05 | 83.55 | 99.15 | 90.15 | 97.70 | 12.45 |
| CRAFT (Ours) | ImageNet | 98.11 | 99.78 | 96.81 | 98.14 | 100.0 | 99.60 | **94.59** | 100.0 | 100.0 | 99.90 | 100.0 | 100.0 | 100.0 | 100.0 | 100.0 |
| | CelebA | 77.85 | **80.16** | 77.63 | 94.53 | 100.0 | 88.50 | 91.57 | 94.30 | 88.40 | 81.00 | 99.60 | 100.0 | 100.0 | 100.0 | 100.0 |
| | LSUN | 96.26 | 99.46 | 98.28 | 95.83 | 100.0 | 96.90 | 99.23 | 100.0 | 99.40 | 99.90 | 99.90 | 100.0 | 100.0 | 100.0 | 98.00 |

### 4.3 GENERATIVE COLLAPSE OF THE DECODER

The final stage translates latent-space chaos into a visual artifact. The VAE decoder, $\mathcal{D}(\cdot)$, is trained to map well-behaved latents from $\mathcal{M}_{real}$ into natural images. When fed a chaotic $x_0$ from a forged sample, the decoding process collapses. This difficulty is particularly pronounced in its transposed convolution layers.

These transposed convolution layers, critical for spatial upsampling within diffusion model decoders, are known to be sensitive to local spatial distributions and can amplify even minor misalignments in the input latent features. Due to their inherent overlapping kernel design, these layers introduce periodic interference patterns that culminate in the distinct honeycomb artifacts observed in the reconstructed forged image (e.g., Figure. 1).

Formally, the process ensures that for a forged input:

$$x_0^{fake} \notin \mathcal{M}_{real} \Rightarrow \hat{x}^{fake} = \mathcal{D}(\tilde{z}^{fake}) \tag{13}$$

This transformation of a statistical anomaly into a visually interpretable failure provides a robust and powerful signal for forgery detection.

## 5 EXPERIMENTS

### 5.1 EXPERIMENTAL SETUP

Our primary evaluation is conducted on the DiffusionForensics dataset Wang et al. (2023a), a comprehensive benchmark specifically designed for AI-generated imagery. This dataset includes a balanced collection of authentic images sourced from well-established datasets such as CelebA-HQ Karras et al. (2018), ImageNet Deng et al. (2009), and LSUN-Bedroom Yu et al. (2015), alongside a diverse array of forged samples generated by a wide spectrum of generative models. The forged content spans multiple generation paradigms, including: GAN-bsed methods such as StyleGAN Karras et al. (2021c) and ProjectedGAN Sauer et al. (2021), diffusion models such as ADM Dhariwal & Nichol (2021), DDPMHo et al. (2020), iDDPM Nichol & Dhariwal (2021), PNDM Liu et al. (2022), Stable Diffusion Rombach et al. (2021), LDM, VQ-Diffusion Gu et al. (2022), and Vision-Language Models(VLMs) including IF Saharia et al. (2022), DALLE-2 Ramesh et al. (2022), and Midjourney. The training and testing sets are disjoint at the generator level.

Table 2: **Ablation of Latent Perturbation and Multi-module Fine-tuning.** We evaluate the individual and combined effects of latent perturbation and multi-module fine-tuning. "Baseline" is trained without adaptation, "Fine-tune only" applies fine-tuning with different reconstruction losses, "Perturbation only" applies latent perturbation without fine-tuning, and "Fine-tune & Perturbation" combines both strategies.

| Setting | Fine-tune | Perturbation | Loss Function | ImageNet / CelebA / LSUN |
|---------|-----------|--------------|---------------|--------------------------|
| **Baseline** | ✗ | ✗ | CE | 51.62 / 33.50 / 53.24 |
| | | | L1 + CE | 96.95 / 97.17 / 81.58 |
| **Fine-tune only** | ✓ | ✗ | L2 + CE | 83.70 / 90.64 / 38.49 |
| | | | CR + CE | 96.17 / 98.02 / 94.00 |
| **Perturbation only** | ✗ | ✓ | CE | 75.00 / 72.22 / 85.32 |
| **Fine-tune & Perturbation** | ✓ | ✓ | CR + CE | **98.11 / 98.14 / 94.59** |

We build our method upon the Stable Diffusion Image Variation Pipeline, leveraging its pretrained model. For DDIM inversion Mokady et al. (2023), we setting 50 denoising steps throughout all experiments. During training, we fine-tune only the final $k = 15$ steps of the DDIM process, including the image encoder, UNet, and VAE decoder, while earlier steps remain frozen. This stage-wise adaptation allows the model to remain expressive while selectively amplifying off-manifold instabilities. For all experiments, we use AdamW as optimizer with a learning rate of $1 \times 10^{-5}$ for decoder, $5 \times 10^{-6}$ for encoder and UNet. We set $m$ to 0.4 and $\delta$ as 0.3. $\lambda_{cls}$ is set to 1. For each configuration, we run the experiment multiple times and report the mean result.

## 5.2 COMPARISON TO EXISTING METHODS

We compare our method, CRAFT, with prior representative forgery detection approaches, including CNN-Det Wang et al. (2020a), DIRE Wang et al. (2023a), and AEROBLADE Ricker et al. (2024). Each method is trained independently on one source dataset and evaluated across all three testing domains, encompassing a wide spectrum of generation paradigms such as GAN-based, diffusion-based, and closed-source proprietary models. More detailed experimental results are provided in the Appendix.

Table 1 summarizes the key accuracy results across domains and generators. CRAFT consistently achieves the best cross-domain generalization, maintaining high detection performance even when the training and testing domain differ significantly in semantics or generation characteristics. For example, when trained on ImageNet, CRAFT reaches 98.11% on ImageNet, 98.14% on CelebA, and 94.59% on LSUN. In contrast, existing methods show severe drops in unseen domains, even performing near chance on cross-domain samples.

In addition to quantitative results, Figure 3 provides qualitative evidence of CRAFT's robustness. Even trained solely on ImageNet, our model reliably reconstructs authentic samples with high fidelity while inducing collapse patterns for forgeries, regardless of the test domain. These structured artifacts act as transferable signals of inauthenticity. This visual consistency further supports the claim that CRAFT captures a domain-agnostic notion of authenticity, enabling effective generalization without requiring any domain-specific adaptation.

These results underscore the robustness of our framework, particularly its ability to amplify discriminative reconstruction discrepancies through latent perturbation and CR loss.

## 6 ABLATION STUDIES

We conduct ablation experiments to dissect the mechanisms behind the induced collapse in forged reconstructions and validate the contributions of each component of our method. Specifically, we aim to answer the following questions: (1) Does the contrastive reconstruction loss truly enforce asymmetric behavior? (2) How important is stage-wise fine-tuning in the final DDIM steps? (3) Is

class-guided latent perturbation necessary? Experiments are conducted on DiffusionForensics with all other variables held constant.

## 6.1 CONTRASTIVE RECONSTRUCTION LOSS

We investigate the impact of different reconstruction loss functions on detection performance, as shown in Table 2. Without fine-tuning, the generator fails to introduce a discriminative signal, resulting in near-random detection accuracy across all domains. Fine-tuning with L1 loss significantly improves accuracy, but tends to degrade the quality of real sample reconstructions, leading to blurring and compromised generalization. L2 loss performs worse, likely due to its higher sensitivity to subtle pixel variations that do not align with perceptual differences.

In contrast, our proposed CR Loss consistently achieves superior accuracy, particularly under cross-domain settings. This validates its effectiveness in preserving authentic content while actively forcing forged reconstructions to collapse. Visual results in Figure 1 further illustrate this behavior, reconstructions with CR loss retain semantic fidelity for real samples, whereas fake samples exhibit structured artifacts, thereby facilitating downstream classification. These results confirm the critical role of the reconstruction objective in shaping discriminative latent-space behavior.

## 6.2 LATENT PERTURBATION

To isolate the contribution of each core component, we ablate our framework's two main mechanisms: latent perturbation and contrastive fine-tuning. As shown in Table 2, we compare three configurations: the baseline model without either component, the model with latent perturbation only (Perturbation only), and our full framework combining both (Fine-tune & Perturbation). The baseline model performs at near-random chance, confirming that standard reconstruction offers no discriminative signal. Applying latent perturbation alone provides a significant performance boost, demonstrating that shifting forged latents off-manifold does create a more separable feature space. However, without a model trained to exploit this shift, the effect is inconsistent and yields suboptimal accuracy, failing to induce the desired structural collapse.

In contrast, the full CRAFT framework, which combines perturbation with our contrastive fine-tuning, achieves the highest performance across all domains. This confirms our core hypothesis: while latent perturbation creates the conditions for failure, it is the contrastive fine-tuning that actively engineers the desired failure mode. The model learns to interpret the off-manifold position of a perturbed forgery as an explicit instruction to collapse its reconstruction. This synergistic relationship is essential for transforming the subtle geometric separation in the latent space into a robust and visually interpretable signal.

# 7 CONCLUSION

In this work, we introduced CRAFT, an AI-generated image detection framework that reframes the problem from passive observation to active interrogation. By weaponizing the internal dynamics of diffusion models, CRAFT uses a synergy of class-guided latent perturbation and a novel contrastive reconstruction loss to achieve a state of asymmetric reconstruction. Our method forces forged samples into a structured, visually explicit collapse while preserving the fidelity of authentic inputs. Extensive experiments validate this approach, demonstrating that CRAFT establishes a new state-of-the-art in both detection accuracy and cross-domain generalization.

Beyond these empirical results, this work introduces a new paradigm for generative model forensics. The "active induced failure" framework moves beyond cataloging static artifacts and provides a methodology for dynamically probing the latent geometry and learned sensitivities of complex models. Future work could involve a more precise quantification of the induced collapse, enabling deeper insights into how different generative models respond to induced instabilities and their relation to the underlying geometry of forged representations. This shift toward active, interrogative techniques points to a promising direction for more robust and interpretable detection.

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

# A  APPENDIX

## A.1  EXTENDED IMPLEMENTATION DETAILS

We detail our training configuration and code structure to facilitate reproducibility:

- **Environment:** PyTorch 2.7, CUDA 12.4, Python 3.10
- **Hardware:** 1× NVIDIA L20 (48GB)
- **Batch size:** 8, and use gradient accumulation over 4 steps, yielding an effective batch size of 32
- **Learning rate:** $1 \times 10^{-5}$ for decoder, $5 \times 10^{-6}$ for encoder and UNet
- **Optimizer:** AdamW Loshchilov & Hutter (2017)

The code for this research will be made publicly available upon the acceptance of this paper.

## A.2  COMPARISON TO EXISTING METHODS

In this section, we present a comprehensive comparison of our method (**CRAFT**) against existing approaches across multiple testing domains and generative methods. Specifically, we evaluate each method on three distinct testing domains: **ImageNet** Deng et al. (2009), **CelebA** Liu et al. (2015), and **LSUN** Yu et al. (2015). For each domain, we report:

- The overall detection performance in terms of Accuracy and Average Precision.
- The per-generator detection accuracy, covering representative generative techniques across three major paradigms: GAN-based (e.g., *StyleGAN Karras et al. (2021b)*), diffusion-based (e.g., *ADM* Dhariwal & Nichol (2021)), and latent variable models (LVMs) such as *DALLE 2* and *IF* Saharia et al. (2022), along with commercial models like *Midjourney*.

The detailed quantitative results are provided in Table 3 and Table 4. Specifically, Table 3 reports both the overall detection performance (in terms of Accuracy/Average Precision) on the ImageNet and CelebA testing domains, as well as the per-generator detection accuracy for each generation technique, including methods based on GANs, diffusion models, and LVMs. It also includes the overall performance on LSUN, enabling a clear horizontal comparison across testing domains.

While Table 4 presents a fine-grained breakdown of detection accuracy on individual generation methods within the LSUN domain, providing further insight into the method-wise effectiveness of each detector under domain shift. Together, these results highlight both the robustness and generalization capability of our method under diverse generative paradigms and cross-domain scenarios.

The results reveal significant differences in domain transferability and robustness to generative diversity. Compared with prior methods such as *CNNDet* Wang et al. (2020b), *DIRE* Wang et al. (2023b), *UFD* Ojha et al. (2023b), *AEROBLADE* Ricker et al. (2024), and *AlignedForensics* Rajan et al. (2025), our method achieves consistently superior performance across all settings. Notably, CRAFT demonstrates strong generalization to unseen domains and high detection accuracy across a broad spectrum of generative models, including GAN-based, diffusion-based, latent-variable-based, and commercial models, confirming its robustness and adaptability.

## A.3  ABLATION ON FINE-TUNING STRATEGY

We conduct a detailed ablation study to dissect the contribution of each backbone component (Image Encoder, UNet, VAE Decoder) to our method's success. We evaluate each configuration's quantitative detection accuracy in Table 5 and its ability to produce the desired qualitative structured collapse for forgeries in Figure 4. Our goal is to validate that the synergy between all three adapted components is essential for optimal performance.

Table 3: **Comparison of detection results on the ImageNet, CelebA, and LSUN datasets.** The table reports the overall detection performance (accuracy and average precision) across all domains, as well as model-wise accuracy on ImageNet and CelebA.

| Method | Training Set | Testing Set | | | | | | | | |
| --- | --- | --- | --- | --- | --- | --- | --- | --- | --- | --- |
| | | ImageNet | | | CelebA | | | | | LSUN |
| | | Total | ADM* | SDv1 | Total | SDv2* | IF | DALLE2 | Mid. | Total |
| CNNDet | ImageNet | 82.67/89.89 | 98.28 | 75.76 | 75.22/82.47 | 91.80 | 63.80 | 87.40 | 78.00 | 70.41/98.33 |
| | CelebA | 35.48/77.36 | 6.540 | 20.28 | 71.00/88.85 | **100.0** | 50.70 | 9.400 | 2.000 | 24.99/95.74 |
| | LSUN | 64.15/85.42 | 75.34 | 46.38 | 74.72/89.76 | 93.40 | 33.80 | 75.00 | 62.00 | 92.61/99.58 |
| DIRE | ImageNet | 81.69/93.63 | 99.60 | 64.11 | 96.44/96.76 | 99.90 | 97.70 | 96.60 | 83.00 | 67.20/98.39 |
| | CelebA | 32.25/77.09 | 3.540 | 13.44 | 71.92/89.20 | **100.0** | 52.90 | 11.40 | 3.000 | 12.01/95.66 |
| | LSUN | 68.66/89.07 | 65.26 | 55.68 | **96.92/98.43** | **100.0** | 97.00 | 87.60 | 95.00 | 93.12/99.65 |
| UFD | ImageNet | 76.27/89.45 | 99.02 | 58.67 | 73.97/80.10 | 96.00 | 84.90 | 56.40 | 48.00 | 69.34/98.28 |
| | CelebA | 35.80/77.94 | 14.56 | 15.67 | 92.36/97.01 | 99.90 | 95.00 | 59.60 | 80.00 | 32.32/96.61 |
| | LSUN | 55.27/83.59 | 80.10 | 23.68 | 65.94/79.97 | 84.60 | 65.70 | 38.20 | 0.000 | 96.91/98.85 |
| AEROB-LADE | training-free | **81.06/85.91** | 66.70 | **94.41** | 91.23/93.37 | 88.50 | **99.85** | 78.73 | 94.73 | 68.65/62.61 |
| Aligned-Forensics | ImageNet | 63.33/87.11 | 3.220 | 76.40 | 27.78/72.22 | 0.000 | 0.000 | 0.000 | 0.000 | 41.62/93.48 |
| | CelebA | 30.99/76.65 | 9.940 | 7.760 | 27.78/72.22 | 0.000 | 0.000 | 0.000 | 0.000 | 13.79/76.65 |
| | LSUN | 60.17/86.57 | 26.10 | 57.60 | 27.81/72.23 | 0.000 | 0.000 | 0.200 | 0.000 | 14.09/95.78 |
| CRAFT (Ours) | ImageNet | **98.11/99.75** | **99.78** | **96.81** | **98.14/99.61** | **100.0** | 99.60 | **99.80** | **100.0** | **94.59/99.98** |
| | CelebA | 77.85/91.94 | 80.16 | 77.63 | 94.53/99.22 | **100.0** | 88.50 | 83.60 | **100.0** | 91.57/99.57 |
| | LSUN | 96.26/98.41 | **99.46** | **98.28** | 95.83/95.93 | **100.0** | 96.90 | **100.0** | **100.0** | **99.23/99.99** |

Table 4: **Comparison of detection results on the LSUN dataset.** The table presents detection performance by **Accuracy(%)** on images generated by specific forgery methods within LSUN, where StyleGAN is abbreviated as SGAN, ProjectedGAN as PGAN, diffusion-StyleGAN as dSGAN, and diffusion-ProjectedGAN as dPGAN, DALLE2 as D2.

| Method | Training Set | Testing on LSUN | | | | | | | | | | | | | | |
|---|---|---|---|---|---|---|---|---|---|---|---|---|---|---|---|---|
| | | SGAN* | PGAN | ADM* | iDDPM* | PNDM* | DDPM | SDv1 | SDv2 | dSGAN | dPGAN | LDM | VQD | IF | D2 | Mid. |
| CNNDet | ImageNet | 92.30 | 43.20 | 50.00 | 77.80 | 82.10 | 54.30 | 58.96 | 99.20 | 92.30 | 35.20 | 86.30 | 98.60 | 99.20 | 94.20 | 98.00 |
| | CelebA | 18.30 | 14.30 | 26.50 | 15.20 | 0.500 | 7.812 | 21.20 | 84.10 | 18.30 | 18.80 | 38.10 | 14.00 | 24.00 | 4.600 | 0.000 |
| | LSUN | 99.10 | 86.10 | 96.90 | 99.80 | 99.90 | 94.92 | 87.28 | 98.10 | 99.10 | 86.20 | 98.00 | 98.20 | **100.0** | 82.20 | 56.00 |
| DIRE | ImageNet | 70.60 | 54.60 | 56.20 | 73.60 | 81.50 | 76.69 | 40.83 | 98.70 | 97.30 | 63.60 | **100.0** | 99.80 | **100.0** | 91.60 | 89.00 |
| | CelebA | 5.200 | 1.200 | 2.500 | 0.700 | 0.000 | 0.391 | 4.320 | 61.10 | 8.600 | 1.100 | 26.40 | 4.200 | 9.800 | 2.800 | 0.000 |
| | LSUN | **100.0** | 97.60 | **100.0** | **100.0** | 99.90 | **100.0** | 87.32 | 84.90 | 96.40 | 97.50 | **100.0** | 99.20 | 99.70 | 76.00 | 13.00 |
| UFD | ImageNet | 66.70 | 80.10 | 37.20 | 81.20 | 90.20 | 74.48 | 78.53 | 18.10 | 80.90 | 73.30 | 19.20 | 97.00 | 31.10 | 48.40 | 16.00 |
| | CelebA | 4.300 | 4.100 | 2.700 | 7.900 | 3.200 | 7.550 | 53.34 | 41.40 | 4.200 | 3.100 | 34.44 | 4.100 | 41.80 | 3.600 | 66.00 |
| | LSUN | **100.0** | **100.0** | 99.20 | **100.0** | **100.0** | 99.74 | **99.44** | 95.80 | **100.0** | **100.0** | 85.10 | **100.0** | 87.30 | 95.20 | 51.00 |
| AEROB-LADE | training-free | 55.75 | 59.65 | 72.45 | 71.05 | 61.55 | 78.85 | 24.81 | 83.55 | 66.25 | 60.05 | 99.15 | 90.15 | 97.70 | 83.53 | 12.45 |
| Aligned-Forensics | ImageNet | 82.00 | 87.50 | 89.60 | 94.40 | 88.70 | 91.93 | 20.36 | 0.000 | 55.50 | 92.10 | 0.000 | 52.60 | 0.000 | 0.000 | 0.000 |
| | CelebA | 2.800 | 1.900 | 0.200 | 0.000 | 0.000 | 0.000 | 23.21 | 0.000 | 1.000 | 3.600 | 0.000 | 0.000 | 0.200 | 0.000 | 0.000 |
| | LSUN | 0.000 | 0.100 | 0.000 | 0.000 | 0.000 | 0.000 | 21.34 | 0.200 | 0.600 | 0.000 | 0.100 | 0.100 | 28.90 | 0.800 | 1.000 |
| CRAFT (Ours) | ImageNet | **100.0** | **100.0** | 98.60 | 99.90 | 99.90 | 99.74 | 85.80 | **100.0** | **100.0** | **100.0** | **100.0** | **100.0** | **100.0** | **100.0** | **100.0** |
| | CelebA | **94.30** | **88.40** | 86.30 | 81.00 | 60.90 | 68.49 | **95.36** | 99.60 | **93.70** | **88.10** | **100.0** | **100.0** | **100.0** | **94.60** | **100.0** |
| | LSUN | **100.0** | 99.40 | 99.46 | 99.90 | 99.90 | 99.48 | 98.28 | 99.90 | **100.0** | 99.60 | **100.0** | **100.0** | **100.0** | **100.0** | **98.00** |

Table 5: **Ablation of Fine-Tuning Strategies Across Model Components.** We evaluate six fine-tuning configurations, where each backbone component (Encoder, UNet, and Decoder) is either frozen (✗) or trainable (✓). Results are reported on three domains. Fine-tuning the Encoder yields the most significant gain in accuracy.

| Configuration | Encoder | UNet | Decoder | ImageNet | CelebA | LSUN |
|---|---|---|---|---|---|---|
| w/o fine-tine | ✗ | ✗ | ✗ | 51.62 | 33.50 | 53.24 |
| Only Enc. | ✓ | ✗ | ✗ | 95.98 | 91.25 | 94.57 |
| Only UNet | ✗ | ✓ | ✗ | 82.15 | 77.28 | 86.36 |
| Only Dec. | ✗ | ✗ | ✓ | 53.74 | 74.39 | 43.50 |
| Enc. + UNet | ✓ | ✓ | ✗ | 97.57 | 96.44 | 84.57 |
| Enc. + Dec. | ✓ | ✗ | ✓ | 98.01 | 93.75 | 92.65 |
| UNet + Dec. | ✗ | ✓ | ✓ | 72.29 | 81.02 | 83.05 |
| Enc. + UNet. + Dec. | ✓ | ✓ | ✓ | **98.11** | **98.14** | **94.59** |

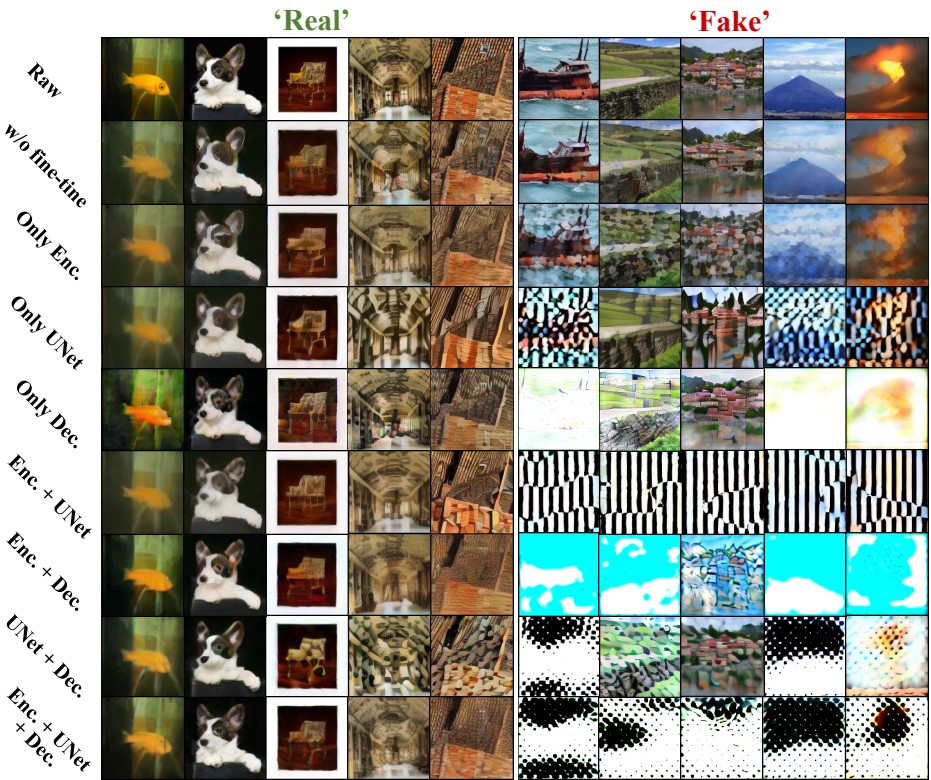

Figure 4: **Visual Impact of Fine-Tuning Different Components.** We visualize the reconstructed outputs of testing sets under various fine-tuning settings. Fine-tuning the UNet alone introduces grid-like artifacts, while adjusting only the Decoder leads to high-frequency distortions. The combination of UNet and Decoder produces a pronounced honeycomb pattern but lacks generalization. Full adaptation of all modules results in both consistent structured collapse and better reconstruction robustness across forgery types.

### A.3.1 ENCODER AS THE KEY TO DETECTION ACCURACY

The results in Table 5 reveal the primary driver of detection accuracy. The baseline model (w/o fine-tune), where all components are frozen, performs near chance level across domain, indicating that adaptation is essential to create a discriminative signal.

Table 6: **Effect of fine-tuning steps under latent perturbation.** We report accuracy (%) on test domains when fine-tuning for $k$ steps. Performance peaks at $k = 15$, indicating that moderate adaptation is optimal for generalization.

| Fine-tuning Steps ($k$) | Testing Dataset | | |
|:---:|:---:|:---:|:---:|
| | ImageNet | CelebA | LSUN |
| 5 | 92.07 | 93.78 | 81.96 |
| 10 | 95.77 | 95.36 | 87.96 |
| 15 | **98.11** | 98.14 | **94.59** |
| 25 | 97.99 | 97.31 | 87.86 |
| 50 | 94.81 | **98.61** | 84.82 |

Among the individual components, fine-tuning the Encoder provides the most significant improvement. For instance, adapting the Encoder alone increases ImageNet accuracy from 51.62% to 95.98%. Furthermore, every configuration involving the adapted Encoder (e.g., Encoder + UNet, Encoder + Decoder) achieves strong performance, confirming its pivotal role. Conversely, the UNet + Decoder configuration, which lacks an adapted encoder, underperforms significantly in terms of accuracy. This quantitatively demonstrates that adapting the Encoder is essential for learning a discriminative feature space that can reliably separate real from fake inputs based on reconstruction dynamics.

These results establish the Encoder as the key contributor to the framework's discriminative power.

### A.3.2 UNet–Decoder Synergy Drives Artifact Emergence

While the Encoder is indispensable for achieving high accuracy, Figure 4 highlights the complementary roles of the UNet and Decoder in generating visually interpretable artifacts that distinguish forgeries from authentic samples.

Fine-tuning the UNet alone causes forgeries to collapse into structured, grid-like noise patterns, suggesting that the denoising trajectory becomes unstable. However, these patterns lack the spatial consistency of the final structured artifact. Fine-tuning the Decoder alone results in a different failure mode, introducing high-frequency and saturated artifacts in forgery reconstructions. This reveals the Decoder's direct control over pixel-level synthesis and its ability to express abnormal latent distributions through visually distinct outputs. The most interpretable and consistent artifact emerges when both the UNet and Decoder are jointly fine-tuned. In this configuration, the UNet produces perturbed latent trajectories, which the adapted Decoder transforms into structured periodic textures. This synergy confirms that artifact emergence is not purely due to one component, but arises from the coordinated instability of the denoising path and the Decoder's learned interpretation of latent anomalies.

### A.3.3 Fine-tuning Steps

Table 6 presents detection accuracy across all domains with increasing fine-tuning steps $k$, under the condition where latent perturbation is enable. A moderate number of updates (e.g., $k = 15$) consistently improves performance across domains, achieving peak accuracy of 98.11% on ImageNet and 94.59% on LSUN. However, overly aggressive fine-tuning deteriorates generalization, especially on LSUN, where accuracy drops to 84.82%. This suggests that moderate adaptation is critical for learning the discriminative failure mode, while excessive fine-tuning lead to overfitting on the spirce domain's artifacts, harming generalization.

### A.3.4 Toward Holistic Adaptation

These findings validate our holistic design. The ablation results present a clear picture: the adapted Encoder is the primary source of high detection accuracy, while the UNet-Decoder synergy is the mechanism for producing the desired generative collapse. Partial adaptations (e.g., Encoder + UNet

or UNet + Decoder) achieving either high accuracy with weak artifact formation, or strong artifact formation with poor generalization.

In contrast, our full model, which fine-tunes all three modules, effectively combines both objectives. The Encoder not only learns a discriminative latent space for robust detection but also preserves reconstruction stability for real samples. Meanwhile, the UNet–Decoder pathway consistently translates off-manifold signals into interpretable collapses in fake reconstructions.

### A.4 Use of Large Language Models

ChatGPT-4.0 was used solely for grammar checking and style refinement to improve the readability of the manuscript.

No large language model was involved in data analysis, experimental design, or the generation of research ideas or results.

