# OpenReview forum: "Triggering Generative Collapse: A Contrastive Inversion Framework for AI-Generated Image Detection"
_ICLR.cc/2026/Conference — ICLR 2026 Conference Withdrawn Submission_

### Official Review · Reviewer_gm4L · 2025-10-25

**Soundness:** 3
**Presentation:** 2
**Contribution:** 2
**Rating:** 2
**Confidence:** 5

**Summary:**

This paper tackles the cross-generalization problem, where the trained detector aims to generalize to various real data distributions and generative models. The paper proposes CRAFT, which tunes the encoder, decoder, and U-Net of the Stable Diffusion model given the real and AI-generated images. The paper proposes two losses: an MLP-based cross-entropy loss and a contrastive reconstruction loss. The paper further aims to strengthen the discrimination between real and AI-generated images via perturbation.

**Strengths:**

1) The paper performs a detailed analysis of each part of the method.

**Weaknesses:**

1) The paper proposes to fine-tune every part of the diffusion model that involves multi-step DDIM sampling. I am not sure how they fine-tune this, since the final output $D_{\theta}(z^{\prime})$ involves a multi-step call of function evaluation (k=15).

2) The evaluation is not done in the standard benchmarks (e.g., GenImage [1], Synthbuster [2], DRCT-2M [3]). Neither is the collection process given to the readers.

3) The baseline is also weak. The authors only compare with the outdated baselines like DIRE or AEROBLADE, of which the latter is training-free. There are a plethora of generalizable AI-generated image detection methods [4,5,6] that emerged recently which should be compared to position the proposed method in the landscape. Furthermore, I strongly suggest that the authors should also compare with the existing diffusion-based methods [3,7] that utilize the reconstruction error.

4) The method can be very slow since it requires tuning not only the encoder, decoder, and the U-Net. Furthermore, the inference speed can be very slow,

5) The contrastive training or utilizing the diffusion model is widely utilized in previous works [3,7,8]. While the proposed method even tune the U-Net, it lacks novelty given that the each component is featured in previous, more efficient methods.


Overall, I think the proposed method is a mixture of existing works that apply diffusion-based methods. Furthermore, I do not find it clear to understand how the proposed method works effectively. Finally, the method fails to compare against existing works in existing benchmark.


[1] GenImage: A Million-Scale Benchmark for Detecting AI-Generated Image, NeurIPS D&B 2023

[2] Synthbuster: Towards Detection of Diffusion Model Generated Images

[3] DRCT: Diffusion Reconstruction Contrastive Training towards Universal Detection of Diffusion Generated Images, ICML 2024

[4] A Bias-Free Training Paradigm for More General AI-generated Image Detection, CVPR 2025

[5] Aligned datasets improve detection of latent diffusion-generated images, ICLR 2025

[6] Improving synthetic image detection towards generalization: An image transformation perspective., KDD 2025

[7] LaRE^2: Latent Reconstruction Error Based Method for Diffusion-Generated Image Detection, CVPR 2024

[8] FIRE: Robust Detection of Diffusion-Generated Images via Frequency-Guided Reconstruction Error

**Questions:**

See Weaknesses

---

### Official Review · Reviewer_JZCC · 2025-10-31

**Soundness:** 3
**Presentation:** 3
**Contribution:** 2
**Rating:** 4
**Confidence:** 4

**Summary:**

This paper proposes a diffusion-based detection framework that integrates latent perturbation with contrastive reconstruction learning to separate real and synthetic images. The approach actively induces divergence during the denoising process, leading to collapse artifacts for fake inputs while maintaining stable reconstruction for real ones. Experimental results on ImageNet, CelebA, and LSUN show promising performance and cross-domain generalization.

**Strengths:**

Strengths
1. The paper proposes a method combining diffusion and contrastive reconstruction, which looks reasonable and makes sense.



2. The method has a certain level of explainability and theoretical grounding, aligning well with the instability analysis of off-manifold denoising.



3. The paper is well structured and easy to follow.

**Weaknesses:**

1. The latent perturbation design assumes a global real-class center, which may be unrealistic for complex and multi-modal datasets (e.g., ImageNet), potentially reducing robustness.

2. The paper lacks an overall motivation: the current innovations (perturbation, contrastive loss, collapse artifacts) appear somewhat disjoint and not presented under a unified theme.

3. Inference requires reverse diffusion + decoding, which is computationally more expensive than direct classifier-based methods, raising concerns for real-time deployment.

4. Experimental comparisons are limited: only simple baselines are included, without evaluation against more recent detection methods. Additionally, in Table 1, the “Training Dataset” column (e.g., ImageNet/CelebA) could cause confusion, since these datasets usually only contain real samples. It should be clarified that both real and fake samples were used in training.

5. Related work is insufficient. Prior research in forgery detection—such as RECCE (End-to-End Reconstruction-Classification Learning for Face Forgery Detection), DCL (Dual contrastive learning for general face forgery detection), and prototype learning approaches（Adaptive Face Forgery Detection in Cross Domain）—has explored similar directions. The proposed method clearly connects to these works, and they should be discussed properly in the related work section.

Overall, the paper draws inspiration from existing work on reconstruction, contrastive, and prototype learning methods, but combines them with diffusion dynamics to amplify off-manifold deviations, which is a reasonable and interesting idea. However, the issues regarding robustness, motivation, limited comparisons, and insufficient related work discussion need to be addressed before this paper can be considered strong.

**Questions:**

1. Hyperparameter Sensitivity Not Analyzed

The method introduces multiple hyperparameters (λ for perturbation, m/δ/γ for loss margins), but only k is ablated (Table 6). How sensitive is performance to λ, m, δ? Were these tuned per dataset or fixed globally? This impacts reproducibility and practical deployment.

2. Honeycomb Artifact Explanation Lacks Rigor

Section 4.3 attributes collapse patterns to "transposed convolution overlapping kernels," but provides no quantitative evidence (e.g., frequency analysis, layer-wise ablation). Why honeycomb specifically, not grid or stripe patterns? This weakens the theoretical contribution.

3. Adversarial Robustness Not Addressed

If an attacker knows the real-class center μ_real and the collapse mechanism, could they craft adversarial fakes that avoid detection (e.g., by optimizing latents toward μ_real)? The paper assumes "naturally generated" fakes but ignores adversarial scenarios.

---

### Official Review · Reviewer_by5m · 2025-11-01

**Soundness:** 2
**Presentation:** 2
**Contribution:** 1
**Rating:** 2
**Confidence:** 4

**Summary:**

The paper proposes an active detection paradigm for AI-generated images that deliberately provokes diffusion models to fail on forgeries while preserving reconstructions for authentic inputs. Concretely, it introduces CRAFT, which (i) applies a class-guided latent perturbation that shifts an inverted latent away from the running “real-class” center, and (ii) fine-tunes only the last DDIM denoising steps with a contrastive reconstruction (CR) loss. The mechanism is supported by an analysis of exploding Jacobian norms during reverse diffusion and the sensitivity of transposed-convolution decoders to perturbed latents. Empirically, on DiffusionForensics across ImageNet, CelebA, and LSUN, CRAFT attains strong cross-domain generalization and SOTA-level accuracies; ablations indicate the encoder is pivotal for detection accuracy, and the UNet decoder synergy produces consistent, interpretable failure patterns.

**Strengths:**

The proposed method demonstrates selective "active" detection by inducing structured latent-space collapse specifically for counterfeit inputs while maintaining stable, high-fidelity reconstructions of genuine images, thereby achieving clear and interpretable discrimination. It further provides valuable mechanistic insights by analyzing Jacobian norms and decoder behaviors, resulting in consistent honeycomb artifacts that serve as human-interpretable signatures rather than relying on opaque numeric scores. Additionally, the approach exhibits strong generalization capabilities across diverse domains, leveraging lightweight fine-tuning of only the final DDIM sampling steps, supported by comprehensive ablation studies that clearly delineate the individual contributions of encoder, UNet, and decoder components, thus facilitating practical integration into existing workflows.

**Weaknesses:**

The presented methodology exhibits several theoretical and methodological limitations that warrant careful reconsideration:

First, the phenomenon of "active collapse" lacks a rigorous continuous-time theoretical grounding, creating dependence on specific discretization solvers. To establish broader applicability and theoretical clarity, it would be necessary to derive explicit sensitivity bounds such as those obtainable through Grönwall inequalities, Lyapunov exponent analyses, or Jacobian spectral controls in the context of the underlying reverse-time stochastic or ordinary differential equations (SDEs/ODEs). Empirical verification across diverse solver families, including reverse-SDE samplers, higher-order integrators, and varying discretization schedules, is also essential. Without such a continuous-time characterization and empirical evidence of solver invariance, the observed collapse risks being a numerical artifact specific to the DDIM discretization, limiting practical robustness and deployment flexibility in real-world production pipelines where solver and step-count adjustments are routine.

Second, the heuristic use of a "real-class center" as a reference point for latent perturbations inadequately formalizes the notion of manifold deviation. Instead, adopting theoretically grounded measures of off-manifold discrepancy such as Stein discrepancies, Fisher divergence proxies, or sliced score residuals and correlating these measures quantitatively with observed collapse magnitudes would offer a more principled framework. Leveraging these theoretically justified diagnostics could facilitate the construction of calibrated, distribution-shift-aware decision thresholds, potentially through conformal or risk-controlling prediction (RCP) frameworks. Such theoretically anchored metrics would provide operational guarantees (e.g., controlled false-positive rates at fixed coverage levels), critical for ensuring auditability, interpretability, and compliance with moderation or forensic policy standards.

Third, the attribution of observed "honeycomb" artifacts to decoder-origin periodicities is asserted without sufficient quantitative substantiation. To rigorously establish these claims, frequency-domain analyses, local-Jacobian linearization studies are needed. Explicitly differentiating decoder-induced periodicities from intrinsic properties of the score-field geometry is crucial. Producing a formal artifact-attribution analysis would significantly enhance both theoretical clarity and practical explainability. The ability to identify and interpret artifact sources under varied encoder-decoder configurations would further benefit stakeholders involved in auditing, compliance, or legal oversight.

Finally, the inherent non-uniqueness of diffusion inversion processes introduces potential vulnerabilities that the current analysis does not adequately address. It would be beneficial to formally characterize conditions under which the proposed collapse statistic remains invariant or bounded across alternative encoders, latent priors, or latent-space reparameterizations. Complementing this formalization with rigorous adversarial robustness evaluations would further elucidate potential vulnerabilities. Furthermore, the incorporation of randomized solver ensembles or controlled noise injection strategies into the system design could mitigate worst-case failure scenarios and reduce maintenance complexity in scenarios characterized by shifting models or evolving content distributions.

**Questions:**

1. Could you provide a rigorous continuous-time analysis based on the underlying reverse-time stochastic or ordinary differential equations (SDEs/ODEs) to formally bound the perturbation amplification  $\vert\vert J \vert \vert$ within the final discretization interval? Furthermore, can you empirically verify that the observed latent-space collapse phenomenon remains stable and consistent across alternative reverse-SDE samplers, higher-order numerical integrators, and varying $\beta_t$ schedules, specifically reporting metrics such as Jacobian spectra or Lyapunov exponents to substantiate solver invariance?

2. Can the heuristic approach relying on the "real-class center" be replaced by a theoretically grounded, quantifiable measure of off-manifold discrepancy (e.g., Stein discrepancy, Fisher-divergence surrogate, or sliced score residual)? If so, can you demonstrate a clear empirical correlation between this manifold-distance measure and the magnitude of latent collapse, and subsequently derive calibrated detection thresholds, potentially leveraging conformal or risk-controlling prediction (RCP) methods, to ensure controlled false-positive rates under moderate distribution shifts?

3. Could you substantiate your attribution of "honeycomb" artifacts through comprehensive frequency-domain and local-linearization analyses, including Fourier-domain gain assessments and singular-value decomposition (SVD) of the decoder Jacobian? Furthermore, can you evaluate the stability of these artifact attributions under variations of the encoder or VAE architecture, ideally providing a formalized "artifact attribution report" that can be utilized by external auditors in conjunction with the collapse score?

4. Given the known non-uniqueness of diffusion-based inversion procedures, could you formally evaluate the invariance or boundedness of the collapse statistic across diverse choices of encoders, latent priors, and latent-space reparameterizations? Additionally, can you systematically assess the robustness of the proposed detection mechanism against counter-forensic interventions and examine the potential benefits of randomized solver ensembles or noise-injection techniques to quantify and mitigate worst-case performance degradation in practical deployments?

---

### Official Review · Reviewer_vvxK · 2025-11-01

**Soundness:** 3
**Presentation:** 3
**Contribution:** 3
**Rating:** 4
**Confidence:** 3

**Summary:**

This paper tackles the problem of AI-generated image detection by actively inducing failures in a generative model. The authors propose a forensics approach called CRAFT (Contrastive Reconstruction Amplification Forensics Technique), which finetunes a diffusion model to produce drastically degraded reconstructions for fake images while maintaining good reconstructions for real images. Specifically, the method introduces a class-guided latent perturbation to push inputs slightly off the real data manifold, and employs a contrastive reconstruction loss during the final denoising stages of a DDIM diffusion model. This causes AI-generated inputs to trigger visible generative collapse artifacts in the reconstructed output while real images remain largely unchanged. Experiments show strong experimental performance over detection and cross domain generalization on different experimental setups.

**Strengths:**

- Interesting idea using contrastive reconstruction loss. This is a fresh viewpoint for reconstruction based AI generated image detection method since this paper introduces significant shift from error-based detection to an active failure induction strategy.
- This paper is well written, organized and easy to read.
- Impressive performance on empirical results.

**Weaknesses:**

- Despite strengths, requiring fake images for training is still critical drawback.
- The core assumption is that AI-generated images lie off the true data manifold and will exhibit reconstruction failures. But this assumption can be violated in real world given the performance of state of the art generative models.
- Concerns on generalization capability since it uses classifier guidance.

**Questions:**

- How are the “authentic-class centers” determined in practice for an input image?
- How would it perform if an input image comes from a very different distribution than the model was trained on?

---

### Note · Authors · 2025-12-02

I have read and agree with the venue's withdrawal policy on behalf of myself and my co-authors.